# Computer Vision Applications and their Ethical Risks in the Global South

Charles-Olivier Dufresne-Camaro*      Fanny Chevalier†      Syed Ishtiaque Ahmed‡

*†‡ Department of Computer Science, † Department of Statistical Sciences, University of Toronto

## ABSTRACT

We present a study of recent advances in computer vision (CV) research for the Global South to identify the main uses of modern CV and its most significant ethical risks in the region. We review 55 research papers and analyze them along three principal dimensions: where the technology was designed, the needs addressed by the technology, and the potential ethical risks arising following deployment. Results suggest: 1) CV is most used in policy planning and surveillance applications, 2) privacy violations is the most likely and most severe risk to arise from modern CV systems designed for the Global South, and 3) researchers from the Global North differ from researchers from the Global South in their uses of CV to solve problems in the Global South. Results of our risk analysis also differ from previous work on CV risk perception in the West, suggesting locality to be a critical component of each risk's importance.

**Index Terms:** General and reference—Document types—Surveys and overviews; Computing methodologies—Artificial intelligence—Computer vision—Computer vision tasks; Social and professional topics—User characteristics; Social and professional topics—Computing / technology policy;

## 1 INTRODUCTION

In recent years, computer vision (CV) systems have become increasingly ubiquitous in many parts of the world, with applications ranging from assisted photography on smartphones, to drone-monitored agriculture. Current popular problems in the field include autonomous driving, affective computing and general activity recognition; each application has the potential to significantly impact the way we live in society.

Computer vision has also been leveraged for more controversial applications, such as sensitive attributes predictors for sexual orientation [80] and body weight [42], *DeepFakes* [78, 81] and automated surveillance systems [53]. In particular, this last application has been shown to have the potential to severely restrict people's privacy and freedom, and compromise security. As one example, we note the case of China and their current use of surveillance systems to identify, track and target Uighurs [53], a minority group primarily of Muslim faith. Similar systems with varying functionalities have also been exported to countries such as Ecuador [54].

While advances in CV have led to considerable technological improvements, they also raise a great deal of ethical issues that cannot be disregarded. We stress that it is critical that the potential societal implications are studied as new technologies are developed. We recognize that this is a particularly difficult endeavour, as one often has very little control over how technology is used downstream. Moreover, recent focus on designing for global markets makes it near impossible to consider all local uses and adaptations: a contribution

*e-mail: camaro@cs.toronto.edu
†e-mail: fanny@cs.toronto.edu
‡e-mail: ishtiaque@cs.toronto.edu

making a positive impact in the Global North may turn out to be a harmful, restrictive one in the Global South [1].

Several groups have come forward in the last decade to raise ethical concerns over certain *smart* systems—CV and general-AI systems alike—with reports of discrimination [14, 15, 44, 48] and privacy violations [26, 50, 58, 62]. One notable example comes from Joy Buolamwini, who exposed discriminatory issues in several face analysis systems [15]. However, very few have explored the more focused general problem of ethics in CV. We believe this to be an important problem as restricting the scope to CV allows for more domain-specific actionable solutions to emerge, while keeping sight of the big-picture problems in the field.

To our knowledge, only two attempts to study this problem have been undertaken in modern CV research [45, 74]. While insightful, these works paint ethical issues in CV with a coarse brush, discarding subtleties of how these risks may vary across different local contexts. Furthermore, the surveyed literature in both work focuses extensively on CV in the West and major tech hubs, raising concerns over the generalization of their findings in other distinct communities. Considering people in the Global South are important users of technology, most notably smartphones, we believe it is of utmost importance to extend this study of CV-driven risks to their communities.

In this work, we conduct a survey of CV research literature with applications in the Global South to gain a better understanding of CV uses and the underlying problems in this area. While studying research literature may provide a weaker understanding of the true problem space compared to studying existing systems in the field, we believe our method forms a strong preliminary step in studying this important problem. In doing so, we aim to provide further guidance to designers in building safe context-appropriate solutions with CV components.

To guide our search in identifying potential risks, we use Skirpan and Yeh's moral compass [74] (Sect. 2). Through our analysis (Sect. 3 -5), we aim to answer the following research questions:

*RQ1.* What are the main applications of modern computer vision when used in the Global South?

*RQ2.* What are the most significant computer vision-driven ethical risks in a context of use in the Global South?

*RQ3.* How do research interests of Global North researchers differ from those of Global South researchers in the context of computer vision research for the Global South?

*RQ4.* How do the risks associated with these technologies when used in the Global South differ from those in the Global North?

It is our belief that while certain CV technologies and algorithms are equally available to people in the North and South, the resulting uses, implications, and risks will differ. Our contributions are fourfold: 1) we present an overview of research where CV technology is deployed or studied in a geographical context within the Global

---

[1] We use the United Nations' M49 Standard [77] to define the Global North (*developed* countries) and the Global South (*developing* countries).

South, and the needs it aims to address; 2) we identify the principal ethical risks at play in recent CV research in this context; 3) we show that research interests in CV applications for the Global South differ between Global South and Global North researchers; and 4) we show that the importance of ethical risks in CV systems differs across contexts. Finally, we also reflect on our findings and propose design considerations to minimize risks in CV systems for the Global South.

## 2 FRAMEWORK

We are aware of only two studies aiming at identifying ethical risks in computer vision. The first, by Lauronen [45], surveys recent CV literature and briefly identifies six themes of ethical issues in CV: espionage, identity theft, malicious attacks, copyright infringement, discrimination, and misinformation.

The second, by Skirpan and Yeh [74], takes the form of a *moral compass* (framework) aimed at guiding CV researchers in protecting the public. In this framework, CV risks fall into five categories: privacy violations, discrimination, security breaches, spoofing and adversarial inputs, and psychological harms. Using those categories, the authors propose 22 risk scenarios based on recent research and real-life events. In addition, they present a likelihood of occurrence risk evaluation along with a preliminary risk perception evaluation based on uncertainty (i.e. how much is known about it) and severity (i.e. how impactful it is).

To guide our search in evaluating CV in a Global South context, we propose using Skirpan and Yeh's framework based on two reasons: 1) we find the set of risk categories to generalize better in non-Western countries, and 2) it allows us to directly compare the results of our risk analysis in the Global South to the results of Skirpan and Yeh's risk analysis in the West [74]. Nevertheless, we emphasize that the risk categories are by no mean exhaustive of every current risk related to modern CV systems. Indeed, the field is still evolving at a considerable rate and new risks may not all fit well into any of those categories, e.g. *DeepFakes*.

To illustrate the scope of each risk category, we present below one risk scenario extracted from the framework [74] along with a brief description of the risk. Note that we make no distinction between deliberate and accidental presences of a risk, e.g. unplanned discrimination caused by the use of biased data to train a CV system.

### Privacy violations

*"Health insurance premium is increased due to inferences from online photos."* — Scenario #1 [74]

Privacy violations are defined as having one's private information collected without the person's consent by a third party via the use of a CV system. This includes data such as body measurements, sexual orientation, income, relationships, travels and activities.

### Discrimination

*"Xenophobia leads police forces to track and target foreigners."* — Scenario #9 [74]

Discrimination occurs when someone is treated unjustly by a CV system due to certain sensitive characteristics that are irrelevant to the situation. Such characteristics include religious beliefs, race, gender and sexual orientation.

### Security breaches

*"Security guard sells footage of public official typing in a password to allow for a classified information leak."* — Scenario #5 [74]

Skirpan and Yeh broadly define security breaches as a malicious actor(s) exploiting vulnerabilities in CV systems to disable them, steal private information or employ them in cyberattacks.

### Spoofing and adversarial inputs

*"Automated public transportation that uses visual verification systems is attacked causing a crash."* — Scenario #8 [74]

In this context, the objective of spoofing and adversarial inputs is to push a CV system to react confidently in an erroneous and harmful manner. Examples include misclassifying people or situations, e.g. recognizing a robber as a bank employee, or misclassifying fraudulent documents as authentic.

### Psychological harms

*"People will not attend protests they agree with due to fear of recognition by cameras and subsequent punishment."* — Scenario #2 [74]

At a high level, any negative change in how people live based on their interactions with CV systems is considered a psychological harm. Unlike the other categories, this risk is more passive and results from longer-term interactions with CV systems.

We stress that the risk scenarios and their analysis were aimed to be general and reflect "normal" life in the Global North. Therefore, they likely do not generalize well to a myriad of areas in the Global South. For example, scenario #1 used to illustrate privacy violations takes for granted that access to health insurance is available and that people have some sort of online presence. It also assumes that a health insurance company has access to enough data on the population to do any kind of reliable inference.

For this reason, we only use the risk categories to guide our analysis. We believe those generalize well in all areas with CV systems, even though they may not cover the full risk space and what constitutes *sensitive* or *private* information may vary across communities [2, 6].

## 3 METHODOLOGY

In the context of this work, we consider a prior work (system) to be **relevant** to our study if it identifies or addresses specific needs of Global South communities through the use of CV techniques. This criterion can be satisfied in multiple ways: direct deployment of a system in the area, use of visual data related to the area, description of local problems related to CV research, and so forth. Works exploring specific problems in more general settings are excluded, e.g. discrimination at large in CV-driven surveillance systems. We exclude from our study systems designed for China, given their current economical situation.

To find relevant research papers, we conducted a search on four digital libraries: ACM Digital Library, IEEE Xplore Digital Library, ScienceDirect and JSTOR. We have chosen these four platforms to maximize our search coverage on both applied research conferences and journals, and more fundamental research publication venues.

We limited our search to research papers published between 2010 and 2019 to focus on recent advances in the field. We included all publication venues as we wished to be inclusive of all CV researchers focusing on problems in the Global South. We consider inclusion to be critical in achieving a thorough understanding of the many different needs addressed by CV researchers, and the ethical risks of CV research in the Global South.

To identify appropriate search queries, we first began with an exploratory search on ACM Digital Library. We defined our search queries as two components: one **CV term** and one **contextual term**, e.g. *"smart camera" AND "Global South"*. To build our initial list of search queries, we selected as CV terms common words and phrases used in CV literature, e.g. "camera", "drone" and "machine vision". As contextual terms, we chose common words referring to HCI4D/ICTD research areas and locations, and ethical

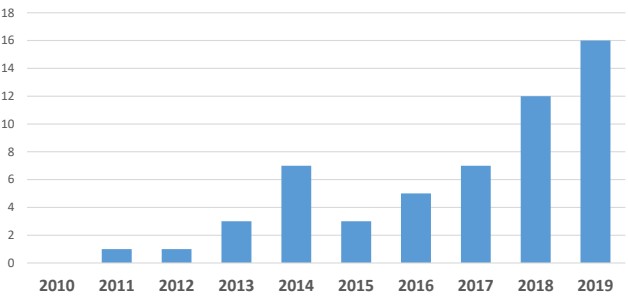

Figure 1: Distribution of publication years for the research papers in our corpus ($n = 55$).

risks in technology , e.g. "privacy", "development", "poverty" and "Bangladesh". Our initial list consisted of 15 CV terms and 24 contextual terms, resulting in 360 search queries.

We selected papers based on their title, looking only at the first 100 results (ranked using ACM DL's relevance criterion). We then discarded terms based on the **relevancy** of the selected papers. Terms that were too general and returned too many irrelevant articles (e.g. CV term "mobile phone" or the contextual term "development") or that continuously returned no relevant content (e.g. CV term "webcam" or contextual term "racism") were discarded.

We retained 5 **CV terms** (*Computer Vision, Drone, Smart Camera, Machine Vision, Face Recognition*) and 10 **contextual terms** (*ICTD, ICT4D, Global South, Surveillance, Ethics, Pakistan, Bangladesh, India, Africa, Ethiopia*) for a total of 50 search queries per platform. This selection allowed us to efficiently cover a wide spectrum of CV research areas (**CV terms**) focused on Global South countries (**contextual terms**). Additionally, this also provided us with an alternative way of covering relevant work by focusing on those considering risks and ethics (**contextual terms**) in CV.

A total of 470 papers were collected over all four platforms using our defined search queries. We then read attentively each abstract and skimmed through each paper to evaluate their relevancy using our aforementioned criterion. Ultimately, 55 unique research papers were deemed relevant.

For each paper, we noted the publication year, the country the CV system was designed for (target location), and the country (researchers' location) and region (Global South or Global North) of the researchers who designed the systems (i.e. the country where the authors' institution is located[2]). Additionally, we noted the main topic of the visual data used by the system (e.g. cars), the application area (i.e. the addressed needs, e.g. healthcare), and the ethical risks at play defined in Sect. 2. For all risks, we only considered likely occurrences, i.e. cases where a system appears to exhibit certain direct risks following normal use.

We did not collect more fine-grained characteristics such as precise deployment areas in our data collection process due to the eclecticism of our corpus: deployments, if they occurred, were not always discussed in detail.

The first author then performed open and axial coding [17] on both the application areas and data topics to extract the principal categories best representing the corpus.

## 4 RESULTS

In this section, we describe the characteristics of the research papers collected ($n = 55$) for our analysis. Table 1 shows an excerpt of our final data collection table used to generate the results below. The complete table is included in the supplemental material. Publication years are shown in Fig. 1.

---

[2]For papers with authors from different countries, we noted the location most shared by the authors.

### 4.1 Location

Researchers' location. Of the 55 articles, 41 were authored by researchers from the Global South and 14 by researchers from the Global North. Specifically, Global South researchers were located in India (20), Bangladesh (10), Pakistan (6), Brazil (1), Iran (1), Iraq (1), Kenya (1), and Malaysia (1). Global North researchers were located in US (11), Germany (1), Japan (1), and South Korea (1).

Target location. For papers authored by Global South researchers, the systems were designed for, or deployed in the same country as the researchers' institutions. For the 14 systems designed by Global North researchers, the target locations were varied: Ethiopia (2), India (2), Kenya (1), Mozambique (1), Pakistan (1), Senegal (1), and Zimbabwe (1). We also note five systems designed to cover broader areas: Global South (4), and Africa (1).

### 4.2 Data topics

Using open and axial coding [17], we extracted six broad categories covering all visual data topics found in our corpus: characters, human, medicine, outdoor scenes, products, and satellite. Each paper is labeled with only one category: the category that fits best with the specific data topic of the paper.

Characters. This category encompasses any visual data that can be used as the input of an optical character recognition system. Systems using text [64], license plates [38, 43], checks [7], and banknotes [55] fit into this category.

Human. Included in this category are any visual data where humans are the main subjects. Such data is typically used in face recognition [12, 39], sign recognition [36, 65], tracking [67], detection [66, 70], and attribute estimation [33]. In our corpus, faces [12, 31, 39] and actions [36, 65, 66, 70] represent the main objects of study using such data.

Medicine. Medical data is tied to the study of people's health. We include in this category any image or video of people with a focus on healthcare-related attributes [47], medical documents [21], and diagnostic test images [20, 22].

Outdoor scenes. Any visual data in which the main focus is on outdoor scenes is grouped under this category. This includes data collected by humans [51], but also cameras mounted on vehicles [11, 18, 79], and surveillance cameras [23, 49]. We however exclude airborne and satellite imagery (i.e. high-altitude, top-down views) from this category.

Products. We group in this category visual data focusing on objects resulting from human labour and fabrication processes such as fruit [30, 68], fish [37, 72], silk [63], and leather [24].

Satellite. Airborne and satellite imagery data such as multi-spectral imagery [59], are grouped under this category.

The distribution of these topics in our corpus is shown in Fig. 2. Fig. 3 shows how this distribution varies with the researchers' region.

### 4.3 Application areas

Through the use of open and axial coding [17], we identified 7 broad application area categories representing all applications found in our corpus: agriculture and fishing, assistive technology, healthcare, policy planning, safety, surveillance, and transportation.

Agriculture and fishing. We include in this category applications related to agriculture, fishing, and industrial uses of goods from these areas. For example, systems were designed to automatically recognize papaya diseases [30], identify fish exposed to heavy metals [37], and estimate crop areas in Ethiopia using satellite data [56].

| Title | Year | Target Location | Researchers' Location (Region) | Data Topic | Application Areas | Risks |
|---|---|---|---|---|---|---|
| Cross border intruder detection in hilly terrain in dark environment [66] | 2016 | India | India (Global South) | Human | Policy plan., Surveillance | Discrimination, Privacy, Spoofing |
| Field evaluation of a camera-based mobile health system in low-resource settings [22] | 2014 | Zimbabwe | United States (Global North) | Medicine | Healthcare | N/A |

Table 1: Excerpt from the final data collection log used in this study. The complete table is included in the supplemental material.

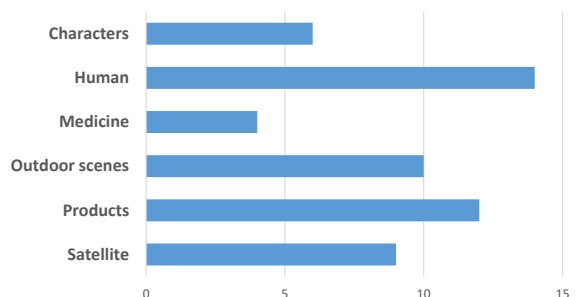

Figure 2: Distribution of the visual data topics ($n = 55$) in our corpus ($n = 55$).

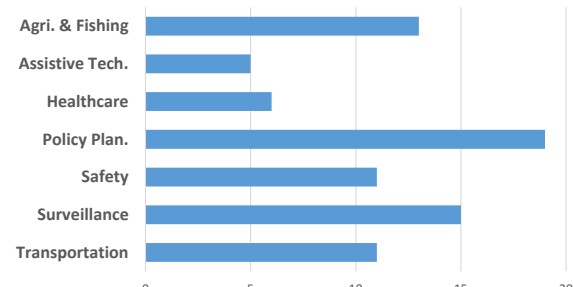

Figure 4: Distribution of application area categories ($n = 80$) in our corpus ($n = 55$).

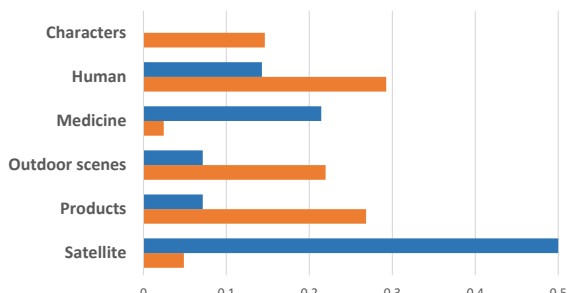

Figure 3: Normalized distributions of data topics for systems designed by researchers from the Global North (blue, $n = 14$) and from the Global South (orange, $n = 41$).

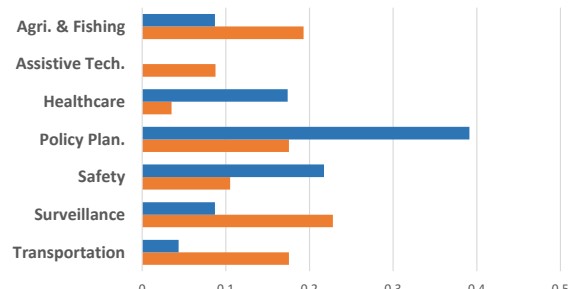

Figure 5: Normalized distributions of application areas for systems designed by researchers from the Global North (blue, $n = 23$) and from the Global South (orange, $n = 57$).

**Assistive technology.** Research papers in this category study the design of CV systems assisting people with disabilities in doing everyday tasks. This includes Bengali sign language recognition systems [36, 65], the *Indian Spontaneous Expression Database* used for emotion recognition [31], and a Bangla currency recognition system to assist people with visual impairments [55].

**Healthcare.** CV research focused on healthcare-related topics fall into this category. We note adaptive games for stroke rehabilitation patients in Pakistan using CV technology to track players [40]. We also include systems used to analyze diagnostic tests [22], and digitize medical forms [21].

**Policy planning.** We include in this category policies tied to urban planning, development planning, environment monitoring, traffic control, and general quality-of-life improvements. We note *SpotGarbage* used to detect garbage in the streets of India [51], a visual pollution detector in Bangladesh [3], and various traffic monitoring systems in Kenya [41] and Bangladesh [23, 60]. We also highlight uses of satellite data to estimate the poorest villages in Kenya to guide cash transfers towards people in need [1], and to gain a better understanding of rural populations of India [34].

**Safety.** In the context of our work, we define safety as the protection of people from general harms. We include in this category CV systems for bank check fraud detection [7], food toxicity detection [37], and informal settlement detection [28].

**Surveillance.** Surveillance applications refer to systems monitoring people or activities to protect a specific group from danger. We make no distinction between explicit and secret acts of surveillance, i.e. being monitored without one's consent. We found systems for theft detection [70], border intrusion detection in India [66], automatic license plate recognition [38], and tracking customers [67].

**Transportation.** CV systems in this category focus on improving and understanding transport conditions in the Global South. The majority of the systems identified in our corpus were aimed at monitoring and reducing traffic [49, 60] and accidents [23]. We also note systems addressing the problem of autonomous driving [11, 79].

| | Agri. & Fishing | Assistive Tech. | Healthcare | Policy Plan. | Safety | Surveillance | Transportation |
|---|---|---|---|---|---|---|---|
| Characters | 0 | 2 | 0 | 0 | 1 | 3 | 3 |
| Human | 0 | 3 | 2 | 2 | 0 | 9 | 0 |
| Medicine | 0 | 0 | 4 | 0 | 0 | 0 | 0 |
| Outdoor scenes | 0 | 0 | 0 | 7 | 4 | 2 | 8 |
| Products | 12 | 0 | 0 | 1 | 1 | 0 | 0 |
| Satellite | 1 | 0 | 0 | 9 | 5 | 1 | 0 |

Figure 6: Co-occurrence frequencies between data topics and application areas in our corpus ($n = 55$).

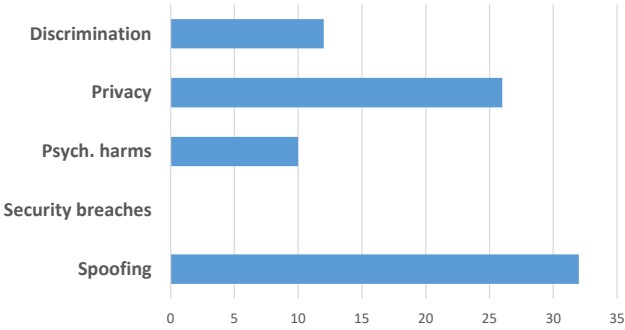

Figure 7: Distribution of risks ($n = 80$) in our corpus ($n = 55$). 18 papers had no identified risks.

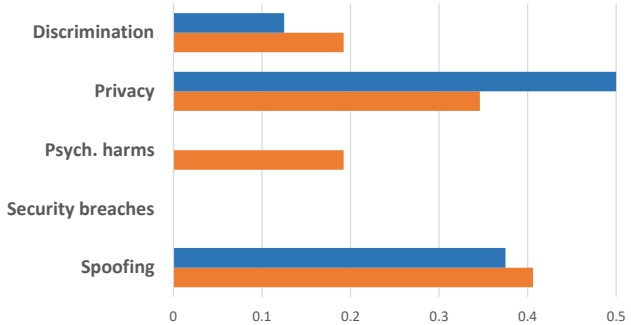

Figure 8: Normalized distributions of risks ($n = 80$) for systems designed by researchers from the Global North (blue, $n = 16$) and from the Global South (orange, $n = 64$).

Each paper was tagged with at least one category and at most three categories (median $= 1$). The distribution of application areas is shown in Fig. 4, followed by the normalized distributions for each researchers' region in Fig. 5. We present the global co-occurrence frequencies between data topics and application areas in Fig. 6.

### 4.4 Risks

We briefly illustrate how we assessed the presence of risks in CV systems (Sect. 3), using two examples from our corpus. Note that we did not consider intent in our analysis; instead, we focused strictly on the possibility of enabling certain risks.

First, we consider a system using satellite imagery designed to identify informal settlements in the Global South, informing NGOs of regions that could benefit the most from their aid. [28]. While the system focuses on areas rather than individuals, it nevertheless collects information on vulnerable communities without their consent (**privacy violations**). Additionally, a malicious actor aware of the system's functionalities could modify settlements to hide them in satellite imagery, e.g. by covering rooftops with different materials (**spoofing**). For these reasons, we tagged the system with two risks.

Second, we examine a sign language digit recognition system [36], which we tagged with no risk. As the system appears to solely rely on hand gestures and focuses on a simple limited task, the presence of all 5 risks appeared minimal.

Fig. 7 shows the overall risk distribution for our corpus. Papers were tagged with at most four risks; 18 had no risks (median $=$ 1). Normalized distributions for each researchers' region appear in Fig. 8. We show the global co-occurrence frequencies between application areas and risks in Fig. 9, and between data topics and risks in Fig. 10. Finally, we present the conditional probability of a risk being present, given the presence of another in Fig. 11.

| | Discrimination | Privacy | Psych. harms | Security breaches | Spoofing |
|---|---|---|---|---|---|
| Agri. & Fishing | 0 | 1 | 0 | 0 | 6 |
| Assistive Tech. | 1 | 1 | 1 | 0 | 2 |
| Healthcare | 0 | 0 | 0 | 0 | 0 |
| Policy Plan. | 4 | 14 | 0 | 0 | 12 |
| Safety | 1 | 7 | 0 | 0 | 8 |
| Surveillance | 9 | 15 | 10 | 0 | 14 |
| Transportation | 0 | 9 | 3 | 0 | 9 |

Figure 9: Co-occurrence frequencies between application areas and risks in our corpus ($n = 55$).

| | Discrimination | Privacy | Psych. harms | Security breaches | Spoofing |
|---|---|---|---|---|---|
| Characters | 0 | 3 | 3 | 0 | 5 |
| Human | 10 | 10 | 7 | 0 | 9 |
| Medicine | 0 | 0 | 0 | 0 | 0 |
| Outdoor scenes | 0 | 6 | 0 | 0 | 8 |
| Products | 0 | 0 | 0 | 0 | 5 |
| Satellite | 2 | 7 | 0 | 0 | 5 |

Figure 10: Co-occurrence frequencies between data topics and risks in our corpus ($n = 55$).

| | Discrimination | Privacy | Psych. harms | Security breaches | Spoofing |
|---|---|---|---|---|---|
| Discrimination | 1 | 0.46 | 0.70 | - | 0.34 |
| Privacy | 1 | 1 | 1 | - | 0.66 |
| Psych. Harms | 0.58 | 0.38 | 1 | - | 0.28 |
| Security breaches | 0 | 0 | 0 | - | 0 |
| Spoofing | 0.92 | 0.81 | 0.90 | - | 1 |

Figure 11: Conditional probability of a risk being present, given the presence of another risk $p(row|column)$ in our corpus ($n = 55$).

## 5 DISCUSSION

Overall, we find that the majority of surveyed CV systems were designed to address specific needs of a community, ranging from food quality control to healthcare, traffic minimization, and improved security. Moreover, we observe that currently popular research areas in the West, such as autonomous vehicles and affective computing, have not achieved the same penetration level so far in the Global South.

We present below our most notable findings and compare them with the findings of Skirpan and Yeh for the Global North [74]. We then present design considerations to minimize each risk, followed by a discussion on the limitations of our study.

### 5.1 Key findings

**CV research for the Global South, by the Global South?**
75% of the papers in our corpus were authored by researchers from the Global South. We find this proportion concerning, as it highlights an ever present risk of colonization of methods and knowledge [75]. This is furthermore important as both the interpretation of visual data and its associated applications are highly dependent on the context in which the data is situated [25]. Thus, we believe this proportion raises an important ethical concern around the interpretation of visual data and the politics tied to the downstream CV applications [35].

Interests differ between regional research groups.
Grouping the distributions of data topics and application areas by researchers' regions (Fig. 3 and 5) reveals significant differences in interests. We note for example the data topic of satellite imagery used in 50% of all Global North systems and only 5% of Global South systems. We speculate that this type of data may be less utilized by Global South researchers as it is typically used to solve larger fundamental problems rather than directly addressing specific needs. Conversely, we find Global North researchers to focus more on data that can be used to solve broader problems.

These distinctions are also visible in application areas: Global North researchers appear to focus much more on applications requiring significant technical expertise, often employing modalities neither aligned with nor originated from local methods [75], with broader ramifications (e.g. policy planning applications identifying at-risk areas through satellite imagery [1, 28]) than applications focused on specific local problems, requiring a thorough understanding of the community and less technical expertise (e.g. applications for transportation, agriculture, and surveillance). In contrast, the regional distribution for Global South researchers is much more uniform, with greater interest in those local problems. Furthermore, these disparities may also be caused by the lack of deeper knowledge about the local problems of the Global South among the CV community in the Global North [35].

Finally, we note that such differences in interests may also be caused by accessibility issues, e.g. data being made available only to certain researchers, high cost to developing new system compared to imports, and computational power issues.

Surveillance applications – promising, but not without risks.
Globally, we find surveillance to be the second most popular application area behind policy planning, an area mostly popular due to its encompassing definition. We attribute this popularity most notably to one factor: surveillance systems are a natural solution to satisfy in part the essential need of security, albeit at the cost of restricting liberties. This trade-off is notably closely tied to the government systems in place, which vary wildly in the Global South [6]. Moreover, we note that the motivations for surveillance vary between cultures, ranging from care, nurture, and protection to control and oppression.

We emphasize that researchers should be especially careful in designing surveillance systems, as we found that most studied here could be easily misused to focus on increasing control over people rather than improving security; we found surveillance to be the application most frequently tied to any risk, and the one with the most diverse set of risks (Fig. 9).

The impact of machine learning on CV risks.
73% of all CV systems analyzed used machine learning techniques to achieve their goals. This has several implications when it comes to the presence of certain risks in CV systems. First, it amplifies the risk of spoofing and adversarial inputs, the most frequent risk in our corpus appearing in nearly all data topics and application areas (Fig. 9 and 10). Indeed, research has shown neural networks to be easily misguided by adversarial inputs [27, 57, 61].

Second, the use of machine learning amplifies the risk of discrimination, in particular the unplanned type, i.e. CV systems using sensitive attributes for decision making without being designed to do so. We found fewer examples of this risk in our corpus as discrimination occurs strictly against people; the risk was mainly found in surveillance and policy planning applications using human data.

Privacy violations – the common denominator.
In our analysis, we found privacy violations to be the second most common risk, far ahead discrimination and psychological harms. Specifically, we found the risk of privacy violations most present in surveillance and policy planning applications, both areas requiring in-depth knowledge of people.

Additionally, we have observed that certain risks were typically present only when privacy violations were too. In particular, the risk of privacy violations was always present when the risks of discrimination and psychological harms were found (Fig. 11). While our analysis does not robustly assess the relations between each risk, this finding suggests an almost hierarchical relation between certain risks, beginning with privacy violations: e.g, for automated discrimination to occur, sensitive attributes of people must first be known (privacy violations). Similarly, the risk of psychological harms depends directly on the presence of other risks. We note however that the risk of spoofing appears to be tied more closely to certain algorithms than risks, although we still expect its frequency to rise as access to sensitive data (privacy violations) increases.

Privacy violations frequency – a likely underestimate.
We note that research on privacy and technology [2,6] has shown privacy to be regionally constructed, differing between cultural regions: e.g. people in the Global South often perceive privacy differently than Western liberal people. Our interpretation of the general definition of privacy violations used in our analysis (Sect. 2) likely does not encompass every cultural definition of privacy in the Global South. As such, there may exist many unforeseen privacy-related risks tied to CV systems, leading us to believe the actual frequency of the risk is even higher than what we have found in our analysis.

Not all risks could be reliably perceived.
We note in our analysis two risks that could not be reliably estimated: security breaches and psychological harms. In fact, we were unable to find a single case of the former. We attribute this primarily to the method used for analysis: detection of the risk would require research papers to present an in-depth description of a system's implementation, highlighting certain vulnerabilities. Similarly, while we have found some systems with risks of psychological harms, we emphasize that we have likely underestimated its actual frequency: detection requires both an in-depth understanding of the system and its subsequent deployments. Moreover, compared to the other four risks, we re-emphasize that the risk of psychological harms results from longer-term interactions and may only become discernible as risk-driven harms begin to occur.

Frequency versus severity.
We first stress that the absence of certain risks does not indicate their non-existence. On the contrary, this informs us to be more wary of these risks as we may have little means to diagnose them in this context. Additionally, we note that the risk frequency estimated here is a different concept than the frequency of harm occurrences tied to the aforementioned risk. For example, while spoofing was found to be the most common risk in our corpus, harm occurrences linked to this risk requires the existence of a knowledgeable malicious actor attacking the system, which may be much more uncommon.

Furthermore, we emphasize that each risk can have severe impacts, regardless of their frequency. For example, while the risk of security breaches may be infrequent, a single occurrence could lead to disastrous consequences, from large-scale information leaks to casualties. Thus, while this study may highlight common pitfalls tied to the design of recent CV systems, we urge designers to consider the potential impacts caused by all risks equally.

## 5.2 Comparison with Global North risk analysis

### 5.2.1 Risk probability

In their work, Skirpan and Yeh [74] identified discrimination, security breaches and psychological harms as the most probable risks. In the present study, we instead identify privacy violations and spoofing as the most probable risks in the Global South. We attribute these differences to two reasons.

First, each work focused on a different body of work. Skirpan and Yeh searched through both research literature and current news, leading to a more diverse set of systems and problems. We, on the other

hand, limited our search to research papers focusing specifically on the Global South to obtain a more grounded understanding. Thus, risks that cannot be reliably estimated from research papers (e.g. security breaches) did not appear frequently in our study. However, this alone is insufficient to suggest the risk space to be different in the Global South: the absence of a risk in research literature does not imply its non-existence in the real world.

Second, we believe these differences to be primarily related to the availability and the acceptance of CV systems in both regions. CV systems in the West, which have become nearly ubiquitous, are often designed to be considerate of the history and politics of the region. However, in the Global South, fewer systems are designed and, in most cases, they involve transferring existing Northern technologies to the Global South. The designed systems thus tend to be less considerate of the culture and politics of the Global South [35], limiting among other things their level of adoption by the community. Moreover, we re-emphasize the pseudo-hierarchical nature of the risks studied here: privacy violations appear to serve as a *springboard* for the other risks to arise. Thus, in areas where CV technology is relatively recent, we should expect to find privacy violations to be more frequent than other risks. This, combined with the extensive use of machine learning techniques, can also explain why the risk of spoofing and adversarial inputs was found to be more frequent than the three risks highlighted by Skirpan and Yeh.

### 5.2.2 Risk Perception

Skirpan and Yeh [74] have also evaluated risks in terms of severity and uncertainty, leading them to identify discrimination and security breaches as the most potential important risks. While we have not formally performed this type of analysis in our work, we speculate the problem space in the Global South to be different.

We believe that at this time, the most important risk is privacy violations, in part due to its gateway property to other risks and the current availability of CV systems in the Global South. We note privacy violations can not only enable automated discrimination in CV systems, but also broader discrimination and security risks. For example, by surveilling and collecting specific sensitive attributes of its people, a state could identify and arrest, displace, or mistreat certain marginal groups [53].

Additionally, our perception aligns with a growing body of HCI4D/ICTD research on the more general problem of privacy and technology [2, 4–6, 32, 73] showing privacy to be regionally constructed and highlighting a general lack of understanding from technology designers, which in turn suggests there may be important privacy (and security) concerns notably at gender and community levels unforeseen in the Global North.

### 5.3 Design considerations for risk minimization

As a general consideration to minimize risks in CV systems, we urge researchers to ensure their designs are context-appropriate for the focused area. Designing such systems requires expertise in technical domains such as CV and design, but also in humanities and social sciences to identify potential risks for a given community. As such, we recommend involving more local researchers in the design process. Aside from risk minimization, the participation of local people is critical in ensuring the goals of the system are aligned with the true needs of the community. We refer designers to Irani et al.'s work on postcolonial computing [35] for more in-depth guidance on this design process. For the rest of this section, we provide design considerations for each ethical risk considered in our study.

**Privacy violations.** We briefly highlight two common privacy-related issues in modern CV systems. First, decision making processes have become increasingly obfuscated: decisions could unknowingly be based on private attributes, rather than the ones expected by the designer and consented by the user. Second, non-users in shared spaces may unconsciously be treated as users. We refer

to a growing body of work on privacy-preserving CV for concrete solutions [16, 69, 71, 76], such as Ren et al.'s *face anonymizer* for action detection. [69].

**Discrimination.** Discrimination in smart CV systems and general AI systems has been extensively researched in recent years [15, 83]. The use of biased, imbalanced, or mis-adapted data and training schemes are typically the cause of this risk. We present two strategies to address these issues. First, datasets can be adapted and balanced by collecting new context-appropriate data, e.g. ensuring that every gender and ethnicity is equally represented in a human dataset [19]. Second, learning schemes can be modified to achieve certain definitions of *fairness* [10, 52, 82]. We refer to Barocas et al.'s book [13] for a more in-depth analysis of general fairness-based solutions to minimize discrimination.

**Spoofing and adversarial inputs.** In general, risk minimization can be achieved by improving system robustness. Solutions include taking into account external conditions that could affect performance (e.g. variable lighting), using sets of robust discriminative features in decision making, and improving measurement's quality. However, all systems are not equally prone to fall to the same types of attacks. For example, systems involving neural networks are typically weaker to precise adversarial inputs [27, 57, 61]. In this context, we refer to recent advances in the field for specific solutions [8, 9, 29, 46].

**Security breaches.** Our considerations here are limited as the minimization of this risk depends heavily on the type of system. We strongly encourage designers to engage with experts in this field to develop robust context-specific solutions. Nevertheless, general solutions include avoiding system deployments in areas with easily accessible sensitive information and limiting access to the system.

**Psychological harms.** We urge designers to engage with the communities and involve local experts to determine what constitutes an acceptable and helpful CV system, and to follow these guidelines during design and deployment. We again emphasize how this risk is closely tied to all four others: any occurrence of harm related to the other risks will inevitably lead people to modify their lifestyle. Thus, the minimization of this risk depends directly on the minimization of the other four risks.

### 5.4 Limitations

**Data collection.** Our analysis does not take into account every use of CV in the Global South. Indeed, we have only considered certain systems that were published as research papers in specific digital libraries and tagged as relevant during our search. We thus only present a partial view of the real practical uses of CV in the Global South, i.e. systems encountered by the public in their daily lives, and the subsequent ethical risks.

Furthermore, we note that our relevance criterion can be interpreted and applied differently. For example, "use of visual data related to the area" could refer to the use of a publicly-available dataset representative of the target location. However, it could also refer to any imagery that appears to be related to the area. Due to the scarcity of such datasets, we opted for the latter definition, which resulted in a more relaxed definition of relevancy.

Finally, we acknowledge that the collected characteristics are for the most part surface-level, in part due to the eclecticism of the corpus: e.g. precise deployment areas (e.g. office spaces) could not be collected uniformly as deployments were not always discussed.

**Risk evaluation.** First, we were unable to reliably assess the presence of the risks of security breaches and psychological harms in our corpus. While we have attempted to mitigate this issue by estimating the risks' frequency based on our understanding of the systems, our evaluation does not likely represent well their actual presence in the Global South.

Second, we note that our corpus was heavily skewed towards certain Global South countries: 69% of the systems were designed for the countries of India, Bangladesh and Pakistan. We thus acknowledge that our results do not represent equally well, and may not generalize well to every community in the Global South.

Finally, we note that our risk frequency and perception analyses directly reflect our views of the problem, namely as designers and researchers from the Global North, which are themselves inspired by previous work [74]. Nevertheless, we believe that our overview of the typical CV-related ethical risks can serve as a basis to assist researchers in designing safe context-appropriate CV systems for the Global South.

## 6 CONCLUSION

We have presented an overview of applications and ethical risks tied to recent advances in CV research for the Global South. Our results indicate that surveillance and policy planning are the most explored application areas. We found research interests to differ notably between regions, with researchers from the North focusing more on broader problems requiring complex technical solutions. Risk-wise, privacy violations appears as the most likely and most severe risk to arise in CV systems. This last result differs from those of Skirpan and Yeh [74], which were obtained by analyzing both research literature and news events with a focus on the Global North. Taken together, our findings suggest that the actual uses of CV and the importance of its associated ethical risks are region-specific, depending heavily on a community's needs, norms, culture and resources.

As future work, this study can be extended to non-research CV applications, e.g. commercial systems, to gain a more thorough understanding of the situation in the Global South. Additionally, similarly to what Skirpan and Yeh proposed in their work [74], we believe the next major step in improving our understanding of these issues is to survey direct and indirect users of CV systems and study their interactions in specific areas of the Global South. Only then can we gain a more precise and representative understanding of the importance and the frequency of each of those ethical risks.

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
