# OpenReview forum: "Computer Vision Applications and their Ethical Risks in the Global South"
_graphicsinterface.org/Graphics_Interface/2020/Conference — GI 2020_

### Official Review · AnonReviewer3 · 2020-02-09
**Review: Overview of Computer Vision Applications and Risks in the Global South**

**Rating:** 7
**Confidence:** 4

**Review:**

The paper presents insights from a literature review of computer vision and risk factors in the global south.

The paper is well written, timely and provides limited, but interesting insights and a good framework to discuss differences in computer vision use between regions.

The limitations are primarily in the approach: A literature review will only reveal research systems published and neglect commercial efforts or research prototypes that were never presented in a publication. I think it is important to understand the work that this limitation is the lens the results need to be interpreted through. The authors express these concerns in the limitation, but the paper would benefit from addressing the limitation of the collected data early and assure that the presented insights are framed accordingly.

Overall, I think the manuscript holds merit and the work would spawn interesting conversation at GI.

---
//Abstract
How did were the papers reviewed selected?
I personally don’t like the use of the acronym throughout the paper. It’s not a large issue, but it decreases readability for a very small gain.


//Introduction
I am not sure if the dimensions mentioned in the abstract (i.e.,”[…] three principal dimensions: where the technology was designed, the addressed problems and the potential ethical risks arising following deployment.”) at the end of the abstract are supposed to line-up. Currently there is some similarity, but the authors might want to align both to strengthen the relevance of the perspective taken.

//Framework
I wonder if literature on ethics in AI in general would be informative for understanding ethics in regard to computer vision specifically. I agree with the use of the frame, but see opportunity for improvement regarding the background literature.

While the scenarios might not apply to the global south, the underlying ethical principle would translate. I find the paragraph 218-229 difficult to parse—the ethical standards should be the same even, but there might just at this moment just not be an applicable scenario in certain regions—a differentiation that also hold in the global north; for example, poverty and a lack of health care in the US might lead to a similar situation.

//Methodology
While I don’t see any immediate flaws with the mythology, it seems that the authors decided to define their own approach instead of following existing guidelines for systemic reviews.

//Discussion
I wonder if not many of the current AI applications that are have potential risk are not on the commercial side. While the insights about research produced is most interesting the real impact on lives using AI and CV is currently made in commercial settings, e.g., Sesame Credit in China.

Further, concluding that just because there were no publications that point to risks of computer vision in certain areas, doesn’t imply that these risk don’t exist, but just that nobody has published on the results. An important distinction that the authors should make clear to avoid confusion by the reader.

---

### Official Review · AnonReviewer1 · 2020-02-12
**Review for: Overview of Computer Vision Applications and Risks in the Global South**

**Rating:** 7
**Confidence:** 3

**Review:**

This survey paper discusses an analysis of the application areas for computer vision technology in the Global South and their potential risks. The authors analyzed 55 research papers using open and axial coding to identify application ideas and used the moral compass framework to categorize the types of risks associated with the CV-related research projects.

Overall I found the paper to be interesting and think it clearly identifies the goal, contributions and limitations of this work. The methodology is well explained and the rationales provided make sense. The informal comparison with results from the Global North was also very interesting to read about. That said, a few things could have been better clarified and the discussion further enhanced. Below are some questions and thoughts I had from reading the paper:

- It would have been good to see a definition of what the authors consider CV systems or what does it encapsulate early on in the paper. For example, would systems that used publically shared media data be considered in the scope of CV systems? I may have missed it, but did not see any references to algorithms that may have caused ethical issues using social media data for example.

- The authors covered several variables such as country, data topic, and application area. I wondered if the authors considered the location of where the technology was deployed, not in terms of country, but more specifically where was it used e.g., offices, streets, home, hospital? This may have helped shed more light on the types of application areas explored and may have provided more context for the risk assessment.

- Results -- currently as written the results are a mix of somewhat obvious or expected results (e.g., all technologies have a potential second, third...n order effect which could be harmful to people) and those that are surprising (e.g., important trends in application areas). Readability can be improved if the paper clearly identified the most surprising results helping the readers easily learn about the takeaways.

- Design considerations -- I found them to be less effective given the nature of the study. I am not sure design considerations are even necessary for this paper.  A rich discussion is perhaps a good conclusion for this paper. Perhaps some of the system work discussion in the design considerations section can be used instead to provide more context for the types of systems the research papers included to help readers better understand how the risk was estimated.

- Minor comment -- it would have been nice to read an explicit motivation for this work.

---

### Official Review · AnonReviewer2 · 2020-02-12

**Rating:** 3
**Confidence:** 3

**Review:**

The paper conducts a survey of computer vision (CV) applications in the global south (i.e. developing countries).  The aim is to identify the main applications of CV and related ethical risks that exist in developing countries, and determine how these applications and ethical risks differ from the same in the global north (i.e. developed countries).

The authors identify 55 research papers and manually code them on the basis of location, data topic, application domain, and ethical risk type.  From this coding, the authors draw conclusions regarding the prevalence of different applications and risk types in the global south vs global north, and speculate as to why these differences emerge.

There is a foundational, and to me somewhat questionable assumption, that a relatively small number (55) of research papers are indicative of the actual use and deployment of CV systems.  Moreover, these papers are selected by the authors especially to be related to the global south, so conclusions contrasting trends in the global south vs the global north seem to be inappropriate.

In addition, the paper's contribution seems to be out of scope of GI -- there is no technical contribution to computer graphics or HCI methods.  It could be of interest to researchers interested in the overlap between policy and computer vision applications, but most likely not to the GI community.

---

### Meta-Review · Area_Chair1 · 2020-02-12

**Recommendation:** Accept
**Confidence:** 3

**Metareview:**

The paper received mixed scores from the reviewers (7, 7, 3). I believe that a survey paper is a valid contribution type for GI's HCI track, as it helps further our knowledge about designs that have been explored and highlights what can be done in the future. While the paper has shortcomings (addressed in the paper) and offers limited results (as acknowledged by all reviewers), the results are timely (R3) and interesting (R1, R3). Based on this, I recommend that the paper be accepted.



Below I summarize the key issues identified by the reviewers and encourage the authors to read through the individual reviews carefully to address other concerns.

 - Address the methodological limitations early in the paper (R2, R3)
 - More directly define the scope of this work (e.g., what does CV-systems encapsulate (R1), acknowledge that risks not identified in the analysis may still exist (R3))
 - Provide more details about background literature (R1, R3)
 - Clearly highlight the surprising and new results (R1)


Recommendation: Accept

---

### Decision · Program_Chairs · 2020-02-18

Accept